# Optimization of Irrigation Scheduling for Improved Irrigation Water Management in Bilate Watershed, Rift Valley, Ethiopia

Kedrala Wabela [1,*], Ali Hammani [1], Taky Abdelilah [1], Sirak Tekleab [2] and Moha El-Ayachi [1]

1 Institute of Agronomy and Veterinary Hassan II, Rabat 10101, Morocco
2 Department of Water Resources and Irrigation Engineering, Hawassa University, Hawassa P.O. Box 5, Ethiopia
* Correspondence: kedruwab@gmail.com; Tel.: +212-677475839 or +251-910185922

**Abstract:** The availability of water for agricultural production is under threat from climate change and rising demands from various sectors. In this paper, a simulation-optimization model for optimizing the irrigation schedule in the Bilate watershed was developed, to save irrigation water and maximize the yield of deficit irrigation. The model integrated the Soil and Water Assessment Tool (SWAT) and an irrigation-scheduling optimization model. The SWAT model was used to simulate crop yield and evapotranspiration. The Jensen crop-water-production function was applied to solve potato and wheat irrigation-scheduling-optimization problems. Results showed that the model can be applied to manage the complicated simulation-optimization irrigation-scheduling problems for potato and wheat. The optimization result indicated that optimizing irrigation-scheduling based on moisture-stress-sensitivity levels can save up to 25.6% of irrigation water in the study area, with insignificant yield-reduction. Furthermore, optimizing deficit-irrigation-scheduling based on moisture-stress-sensitivity levels can maximize the yield of potato and wheat by up to 25% and 34%, respectively. The model developed in this study can provide technical support for effective irrigation-scheduling to save irrigation water and maximize yield production.

**Keywords:** SWAT model; optimization; simulation; irrigation-scheduling; potato; wheat

## 1. Introduction

Agriculture, which uses approximately 70% of the world's freshwater withdrawals for irrigation, is the largest consumer of water resources globally [1].The influence of climate change and an increasing demand for water from different sectors affect water availability for agricultural production [2–4]. Moreover, the projected increase in the rate of world-population growth highlights the impending rise in food demand, which will immediately affect farming water-use [3].

Ethiopia is dominantly reliant on agriculture for sources of food and employment. The sector plays an important role, especially for smallholder farmers, who produce 95% of the total agricultural production in the country [5]. Agriculture is a cornerstone of the community in the Ethiopian Rift Valley Lakes Basin (RVLB), as a source of food and income generation. However, the production in the basin has been impacted by climate change and frequent droughts [6,7]. The Bilate watershed, which is situated in RVLB, is vulnerable to climate change, and the availability of water resources in the watershed is deteriorating [7]. Due to climate change, the watershed has experienced a significant drop in rainfall amount and a rise in temperature over the last three decades. Consequently, the stream flow of the Bilate River, which is a source of water for several large, medium, and small-scale irrigation schemes, has been declining [8]. Therefore, in order to cope with the scarcity of irrigation-water sources in those areas, it will be important to practice efficient irrigation-water management techniques such as irrigation-scheduling optimization [9–12].

Agro-hydrological simulation models are capable of illuminating the dynamics of crop growth under different irrigation schedules and climatic conditions. These simulation

models can be used to conduct scenario analysis in order to look for the most effective management approaches [12]. For example, Geerts et al. [13] applied the Aquacrop simulation model to identify the optimal time interval for irrigation-water application, to evade drought stress and attain maximum water-productivity. Li et al. [12] used a soil-water-balance simulation model to study optimal irrigation-scheduling for maize in an arid region of northern China. The Soil and Water Assessment Tool (SWAT) model is a semi-distributed and physically-based simulation model, and it is popular in the simulation of basin-level hydrological processes [14]. It is used to model basin-level hydrology, crop growth, the scheduling of agricultural operations, and climate-change scenarios [15]. SWAT can simulate the effects of various irrigation-water management approaches on crop growth, yield, and hydrological processes. Fu et al. [16] applied the SWAT model to determine optimal irrigation-scheduling for corn and soybean in dryland regions. Sun and Ren [17] used the SWAT model to assess crop yield, and crop-water productivity, and to look at an irrigation-scheduling approach that is optimal for the production of winter wheat and summer maize.

The simulation models can describe the impacts of irrigation-scheduling on yield and crop growth, but they can only answer the question "What if?" [18]. This indicates that more effective irrigation-scheduling depends on scenario analysis of a number of user-decision-based alternatives. In this situation, determining the most effective scheduling approach is dependent on assessments of simulated yield or water productivity. The chosen irrigation schedule, although possibly the best among the options, is probably not the exact optimal global irrigation-schedule [19]. The optimum global irrigation-schedule can be attained by combining the simulation- and optimization-models [20,21].

Optimization of irrigation-scheduling is an important approach for saving irrigation water, improving the productivity of water, and enhancing the benefits to farmers [10,17,22]. Irrigation-scheduling optimization is very helpful to achieve a fair distribution of irrigation water among users at basin level, and it can also improve water-use efficiency. During the application of optimization methods, the irrigation system is defined by creating a sequence of mathematical equations, and the optimal solutions can be determined using optimization-solution technologies [4]. Information on the yield response of crops to water conditions has been required in order to apply scheduling optimization [10]. The crop-water production function describes the association between the crop water used and the yield produced. These associations are complex, since they must involve the impacts of crop moisture-stress at different growth stages [23].

The genetic algorithm (GA), which has been introduced since 1970s [24], is an extensively used algorithm to optimize irrigation-scheduling. It is a search algorism which follows the procedure of natural genetics and selection, which combines the idea of survival of the fittest with genetic operators, to form a strong searching-mechanism. GA solutions are based on parameter coding, searching from a population of points (strings) rather than a single point, and relying on objective-function information rather than auxiliary knowledge [25]. Selection, crossover, and mutation are the three important processes in GA that operate strings and advance to the next generation. GA is found to be useful in the application of irrigation-scheduling optimization, and it has been widely applied to solving simulation-optimization problems [26–28]. Taking into account the thoughts above, the objective of this study is to develop a simulation-optimization model for potato-and wheat-crop irrigation-scheduling for saving irrigation water and maximizing yield. The model will integrate a SWAT crop-growth simulation model and the irrigation-scheduling optimization model outlined to maximize crop yield.

## 2. Materials and Methods

### 2.1. Description of the Study Area

The Bilate watershed is known for its high population density in Ethiopia. Approximately 500 people live in a 1 km$^2$ area [29]. Geographically, the watershed is located between the latitudes of 6°38′18″ and 8°6′57″ N, and in the longitudes of 37°47′6″ and

38°20′14″ E (Figure 1). The watershed has a total area of 5518 km$^2$, with a stream length of 197 km and an elevation range of 1176 m to 3328 m.a.s.l [29]. However, for this study, the watershed area was delineated as 5233 km$^2$. The initial drainage of the Bilate River starts from the Gurage highlands, passes through Siltie, Hadiya, and Kambata, and ends at Abaya Lake, which is one of the largest lakes in the RVLB. The climate in the Bilate watershed is humid and semi-arid, with bimodal rainfall-patterns [30]. The main rainy season is usually the summer monsoon, from June to August [31]. The meteorological data indicated that the mean annual-rainfall ranges from 560 mm in the rifts to 1300 mm in areas of the highlands. The average minimum and maximum temperatures are 16 and 30 degrees Celsius, respectively. The watershed is part of the western rift-margin, which is characterized by deep and wide valleys with several streams. [32]. More than 82% of the land-use type is occupied by agricultural activities (Figure 2a). Nitosols, Cambisols, Vertisols and Leptosils are the major soil-groups in the watershed (Figure 2b).

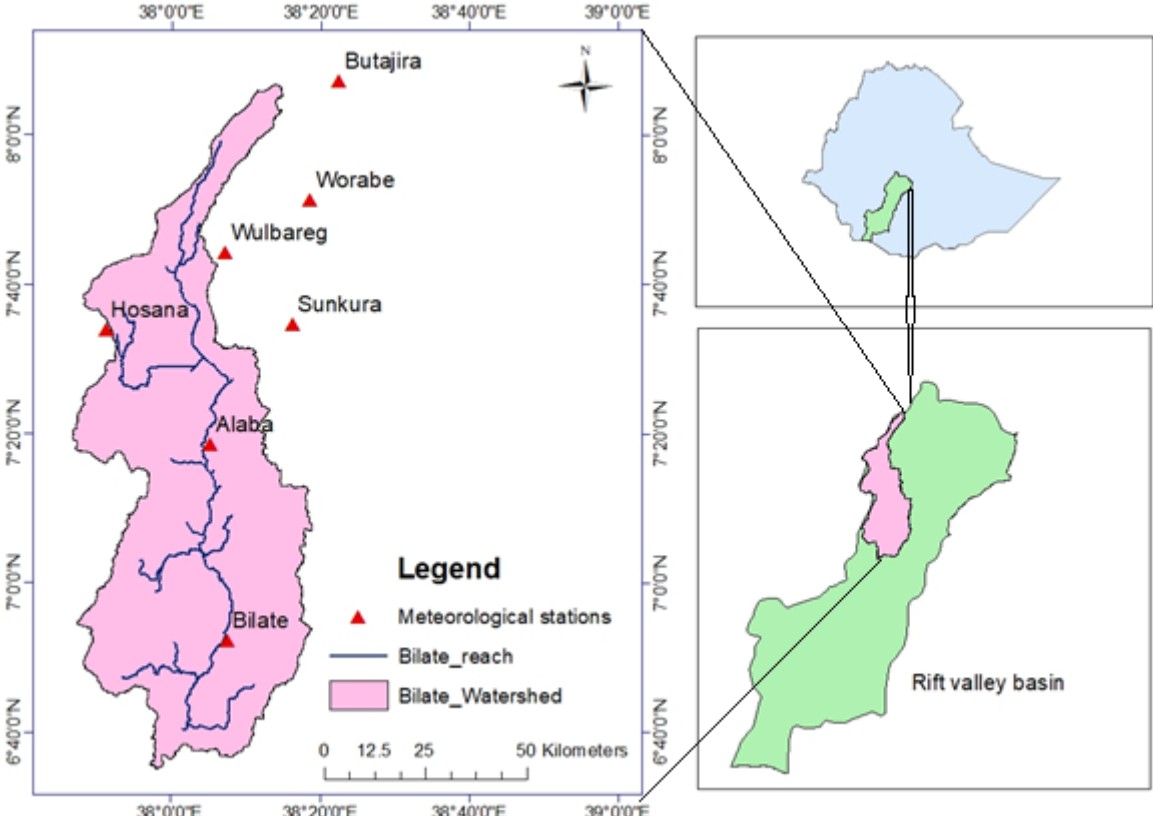

**Figure 1.** Location map of the study area.

## 2.2. Available Data

Temporal- and spatial-data were collected, to establish a SWAT model in the watershed (Table 1). In addition, a field survey was conducted to compile information on the study area's more significant irrigation-crops and current irrigation-production scenarios. The irrigation departments in the districts were contacted for additional crop-production data, seasonal-crop yield, and other necessary details.

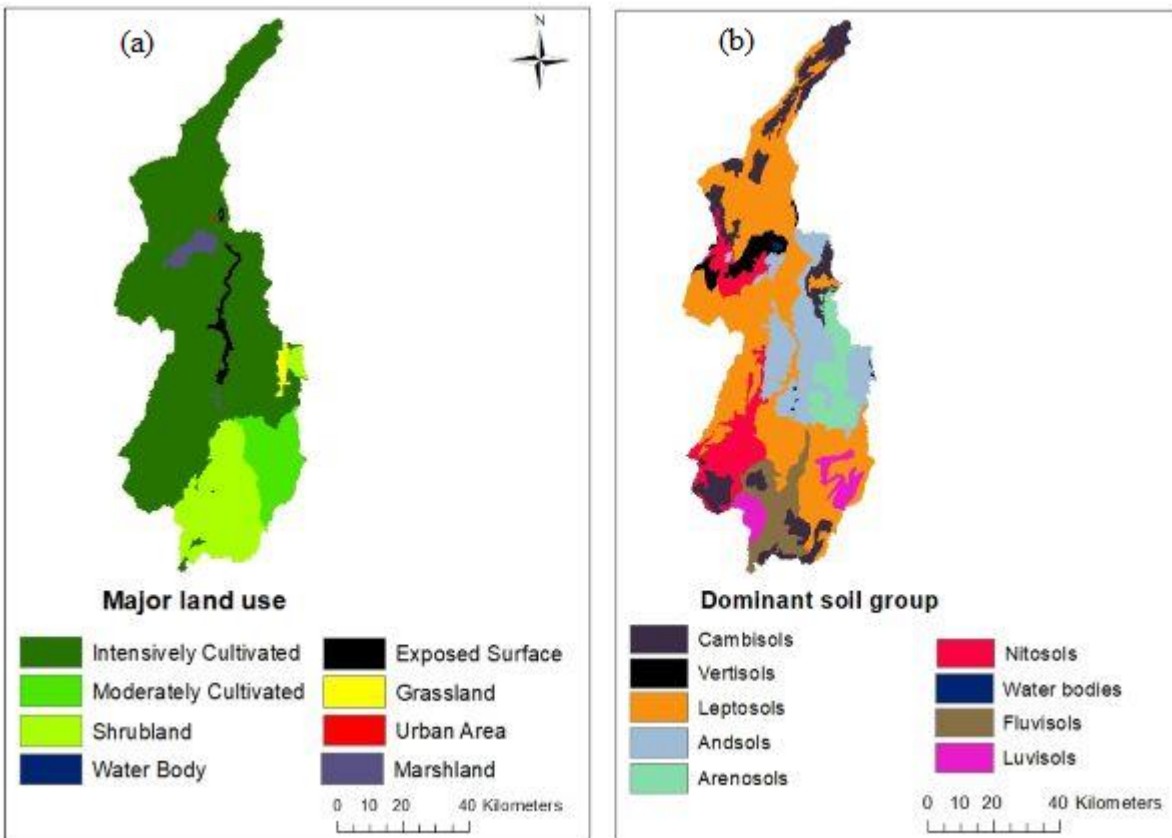

**Figure 2.** (**a**) Land use. (**b**) Dominant soil-group.

**Table 1.** Collected data.

| Data Type | Data Source | Resolution | |
| --- | --- | --- | --- |
| | | **Temporal** | **Spatial** |
| Streamflow data | MoWE | Daily (1991–2008) | - |
| Climatic data | NMSA | Daily (1991–2014) | - |
| Crop data | Zones and districts | Annual (2001–2014) | - |
| Soil data | MoWE | - | 30 m × 30 m |
| Land use and land cover | MoWE | - | 30 m × 30 m |
| Digital Elevation Model (DEM) | USGS | - | 30 m × 30 m |

Note(s): MoWE: Ministry of Water and Energy; NMSA: National Meteorological Service Agency; USGS: United States Geological Survey.

### 2.3. SWAT Model

SWAT is a time-continuous simulation model that can be applied to estimate how land management affects water, agricultural chemicals, and sediment, at the basin level. SWAT divides the basin into sub-basins, which are then further subdivided into pieces of units called hydrologic response units (HRU) [14]. An HRU describes a collection of similar land use and soil types, and it is the smallest unit in a basin. Water resources and agricultural management, and climate, are the main components of the SWAT model.

In this study, the SWAT model was built using a digital elevation model (DEM), climatic data, land-use/land-cover data, and soil data of the study area. The SCS curve number approach (USDA Soil Conservation Service) was applied, to simulate the surface runoff. The Penman–Monteith technique [33] was used to calculate the potential evapotranspiration (PET) and reference evapotranspiration (ETo). SWAT applies the simplified environmental policy integration calculator (EPIC) crop model [34], to calculate plant growth. It uses the above-ground biomass and harvest-index information to determine

the crop yield on the day of harvest. The governing equation for the SWAT model is the water-balance equation, given by

$$SW_t = SW_0 + \sum_{i=1}^{t} \left( R_{day} - Q_{surf} - E_a - W_{seep} - Q_{gw} \right) \tag{1}$$

where $SW_t$ is the amount of soil water-content in mm at time t (day), $SW_0$ is the initial soil water-content on day 1, in mm, $R_{day}$ is the daily rainfall on the i-th day, in mm, $Q_{surf}$ is surface discharge on the i-th day, in mm, $E_a$ is the actual evapotranspiration on the i-th day, in mm, $W_{seep}$ is the amount of water that enters the unsaturated zone on the i-th day, in mm, and $Q_{gw}$ is the amount of return flow on the i-th day, in mm.

Eighteen years of monthly stream flow and fourteen years of annual crop-yield data were used to calibrate stream flow and crop parameters, respectively. Stream-flow parameters were selected from various sources, and their ranges of parameters were fixed. The calibration process becomes more complex if the number of parameters for calibration is extensive, due to the huge number of processes being taken in account [35]. To reduce the complexity, the sensitive stream-flow parameters were identified, based on one-at-a-time (OAT) and global sensitivity-analysis methods [36]. Sensitivity analysis is the process of evaluating the impact of an input change on the output of a model [14]. The t-stat and *p*-values were used to select parameters for each simulation in the sensitivity analysis [37]. For stream-flow parameter-calibration, the program called Sequential Uncertainty Fitting-II (SUFI-2) was applied in the SWAT-CUP. SWAT-CUP provides numerous objective functions with their specified properties used for calibration. The validation process was carried out with independent observed stream-flow data on the same parameters and parameter ranges, in order to be confident in the calibration. Crop parameters were calibrated manually, using annual crop-yield data. Changes in crop-growth parameters and growth constraints such as nutrient stress and water stress were used to simulate actual crop-growth [38]. The crop parameters that have an influence on yield and ETc were identified by changing each parameter's value, one at a time. The calibration process was carried out for several iterations until the change in output value reached an insignificant level.

The model performance was evaluated based on statistical values, including the coefficient of determination ($R^2$), the ratio of mean-squared-error to the standard deviation of the observed data (RSR), and the Nash–Sutcliffe coefficient ($E_{NS}$). The $R^2$ value describes the association between measure- and simulated-data, and its value is between 0 and 1. A value closer to 1 postulates the good model-performance, while a value of less than 0.6 reveals that the model has poorly performed. The value of the $E_{NS}$ ranges from $-\infty$ to 1, and it enumerates how to fit the simulated output to the observed data. It shows how the magnitude of the measured data varies, compared with simulated data. The performance was evaluated based on recommendations given by [39]:

$$R^2 = \frac{\left[ \sum_{i=1}^{n} (Q_o - Q_{oavr})(Q_s - Q_{savr}) \right]^2}{\sum_{i=1}^{n} (Q_o - Q_{oavr})^2 \sum_{i=1}^{n} (Q_s - Q_{savr})^2} \tag{2}$$

$$E_{NS} = 1 - \left[ \frac{\sum_{i=1}^{n} (Q_O - Q_s)^2}{\sum_{i=1}^{n} (Q_o - Q_{savr})^2} \right] \tag{3}$$

$$RSR = \frac{\sqrt{\sum_{i=1}^{n} (Q_o - Q_s)^2}}{\sqrt{\sum_{i=1}^{n} (Q_o - Q_{oavr})^2}} \tag{4}$$

where n denotes the number of observed values, $Q_o$ represents observed discharge-data ($m^3/s$), $Q_s$ represents simulated discharge-data ($m^3/s$) and $Q_{oavr}$ and $Q_{savr}$ represent the average observed- and simulated-values ($m^3/s$), respectively.

### 2.4. Coupling Degree among ETc and Effectivie Rainfall in Irrigation Season

The coupling degree among crop water-requirement ($ET_c$) and effective rainfall ($P_e$) describes how much the effective rainfall satisfied the crop water-demand in the specific growth stages. Information on the extent of $P_e$ to fulfill $ET_c$ in the specific growth stage is beneficial to setting efficient irrigation-scheduling [40]. $ET_c$ depends on the crop coefficient ($K_c$) and reference evapotranspiration ($ET_o$) [41,42]. In this study, $ET_c$ was calculated as follows:

$$ET_c = K_c * ET_o \tag{5}$$

The value of $K_c$ depends basically on the characteristics of each crop and its stage of growth and canopy dynamics. In this study area, $K_c$ values for potato and wheat have not been determined yet. Therefore, to compute $ET_c$, the $K_c$ values from FAO Irrigation and Drainage Paper No. 56 were used. $ET_o$ is the evaporative capacity of the atmosphere, independently of crop type, crop growth-stage, and management conditions, and it is given by

$$ET_o = \frac{0.408\Delta(R_n - G) + Y\frac{900}{T+273}u_2(e_s - e_a)}{\Delta + Y(1 + 0.34u_2)} \tag{6}$$

where $R_n$ is net radiation at the crop surface [MJ m$^{-2}$ day$^{-1}$], G is soil-heat-flux density [MJ m$^{-2}$ day$^{-1}$], T is mean daily air-temperature at 2 m height [°C], $u_2$ is wind speed at 2 m height [m s$^{-1}$], $e_s$ is saturation vapor-pressure [kPa], $e_a$ is actual vapor-pressure [kPa], $e_s$-$e_a$ is saturation vapor-pressure deficit [kPa], $\Delta$ is slope of the vapor-pressure curve [kPa °C$^{-1}$], and $\gamma$ is psychrometric constant [kPa °C$^{-1}$].

$P_e$ is the portion of the rainfall that is actually stored in the soil. It is the difference between total rainfall and actual evapotranspiration. The climatic variables can be used to directly calculate $P_e$. There are several methods to calculate $P_e$. In this study, the USDA Soil Conservation Service method was applied:

$$P_e = \begin{cases} \frac{P(125 - 0.2 * P)}{125}, & P \leq 250 \text{ mm} \\ 125 + 0.1 * P, & P > 250 \text{ mm} \end{cases} \tag{7}$$

where P is total rainfall.

The value of the coupling degree is between 0 and 1, and it is computed as

$$L_i = \begin{cases} 0 & Pe_i = 0 \\ \frac{Pe_i}{ETc_i} & Pe_i < ETc_i \\ 1 & Pe_i \geq ETc_i \end{cases} \tag{8}$$

where $L_i$ is the coupling degree between $ET_c$ and $P_e$ at growth stage i.

### 2.5. Deficit Irrigation-Scheduling

One of the water-resource-management options modeled by SWAT is irrigation operation. The main purpose of irrigation operation is to evaluate the effect of irrigation-scheduling on irrigation systems, crop growth, and yield. Irrigation in an HRU can be scheduled by the user (pre-defined schedule) or automatically by SWAT, in response to a water deficit in the soil [15]. In this study, irrigation-scheduling scenarios were set on the SWAT model, using pre-defined scheduling operations. The timing and depth of the applied water were filled in by the management module. SWAT enables the scheduling of management operations by day or by the fraction of potential heat units. The model examines whether a month and day have been specified for the timing of each operation, before proceeding. In this study, the irrigation-scheduling in the management operation was carried out using a schedule, by day. Eight irrigation treatments (one full-irrigation and seven deficit-irrigation treatments) were used to simulate potato and wheat yield and evapotranspiration. The deficit amount was defined based on the calculated ETc at specific growth-stages. The irrigation depth based on the ratio or percentage in Table 2 was filled

in using the SWAT model at specific growth-stages. The accumulated potential-heat-unit resets to zero at each calendar year. Therefore, to keep going with the calendar-year heat unit, the planting date was set to January 1st. Generally, four operations were scheduled: planting time, fertilization time, irrigation depth and time, and kill/harvest time. During simulation, irrigation efficiency (IRR_EFM) and surface runoff (IRR_SQ) were considered as 70% and 10%, respectively. An auto-fertilizer operation was chosen to replenish the soil nutrients.

**Table 2.** Irrigation-scheduling treatments: irrigation-depth percentage based on $ET_c$.

| | Growth-Stage-Based Deficit Irrigation (% of $ET_c$) | | | | | | | |
| | Growth-Stage Irrigation Depth for Potato (%) | | | | Growth-Stage Irrigation Depth for Wheat (%) | | | |
| TRT | Seedling | Vege.t | Starch ac. | Maturity | Seedling | Vege.t | Grain fill. | Maturity |
|---|---|---|---|---|---|---|---|---|
| CK | 100 | 100 | 100 | 100 | 100 | 100 | 100 | 100 |
| T1 | 25 | 100 | 100 | 100 | 25 | 100 | 100 | 100 |
| T2 | 100 | 25 | 100 | 100 | 100 | 25 | 100 | 100 |
| T3 | 100 | 100 | 25 | 100 | 100 | 100 | 25 | 100 |
| T4 | 100 | 100 | 100 | 25 | 100 | 100 | 100 | 25 |
| T5 | 25 | 25 | 25 | 25 | 25 | 25 | 25 | 25 |
| T6 | 50 | 50 | 50 | 50 | 50 | 50 | 50 | 50 |
| T7 | 75 | 75 | 75 | 75 | 75 | 75 | 75 | 75 |

Note(s): TRT: Treatment; CK: Full-irrigation treatment; Vege.t: Vegetative; Grain fill.: Grain filling; Starch ac: starch accumulation.

### 2.6. Crop-Water Production Function

The association between applied irrigation-water during specific seasons and crop yield is described by the crop-water production function. An alternative definition of the production function that specifies seasonal evapotranspiration as the independent variable rather than applied irrigation-water has been put forth by some agronomic studies [43–45]. There are two principles of crop-water production function [46]. The first one is the "Boule principle," which expresses the multiplicative effect of moisture deficiency on yield, which occurs during different growth-stages [47,48]. The other one is the "arithmetic principle," which defines the additive effect of the water deficiency on yield, which occurs at the different growth-stages [49,50]. Earlier studies on different crops indicated that the prediction ability of the Jensen model was better than other models [44,51]. Thus, in this study, we applied the Jensen model to compute the crop-water-production functions of potato and wheat in the study area. The model is given by:

$$\frac{Y_a}{Y_m} = \prod_{i=1}^{n} \left( \frac{ET_a}{ET_m} \right)_i^{\lambda_i} \tag{9}$$

where $Y_a$ and $Y_m$ are actual and maximum yield from the deficit- and full-irrigation treatment, respectively, (kg/hm$^2$), $ET_a$ and $ET_m$ are the actual and maximum evapotranspiration from the deficit- and full-irrigation treatment respectively, (mm), i represents growth stages, n represents growth-stage number and $\lambda$ is the Jensen's moisture-sensitivity index.

After simulation of yield and evapotranspiration, the Jensen moisture stress sensitivity index ($\lambda$) was calculated for both crops using the Python/Jupyter notebook packages based on a multiple nonlinear-regression analysis.

### 2.7. Irrigation-Scheduling Optimization Model

Optimization of irrigation-scheduling between irrigation cycles for maximum yield was modeled, using the computed crop-water-production function. The seasonal relative evapotranspiration of the deficit-irrigation treatments and the number of days in each irrigation interval were used for the maximization model. During a field survey, irrigation interval- days and duration of irrigation-time data were collected from sample irrigation

schemes in the study area. The calculated moisture stress-sensitivity index was transformed into corresponding irrigation interval-days, using the cumulative curve of the sensitivity index. The optimal relative evapotranspiration for maximum relative yield was calculated by using the genetic algorism (GA) on the platform of MATLAB (R2020a, MathWorks Inc., Natick, MA, USA). The irrigation-scheduling optimization model is

$$\text{Max } Y_a = Y_m * \prod_{i=1}^{n} \left( \frac{ET_a}{ET_m} \right)_i^{\lambda_i} \tag{10}$$

$$\text{Subject to : } \sum_{i=1}^{n} \frac{ET_{ai}}{ET_{mi}} * d - C * \sum_{i=1}^{n} d \tag{11}$$

$$0 < \frac{ET_{ai}}{ET_{mi}} \leq 1$$

where i represents the irrigation interval considered, n represents number of irrigation cycle in the growing season, d is the number of days in one irrigation cycle, and C is the seasonal relative evapotranspiration of the deficit treatments.

### 2.8. General Framework of the Study

The general outline of the study is shown in Figure 3. First, a SWAT model was developed in the study area using DEM, climatic, soil, and land-use data. Next, the SWAT model was calibrated and validated with stream-flow- and crop-yield data. In the calibrated SWAT model, full- and deficit-irrigation treatments were scheduled to simulate potato and wheat yields and evapotranspiration. The Jensen moisture stress-sensitivity index was then computed from simulated yield and evapotranspiration, and the Jensen crop-water-production function was developed for potato and wheat. The optimal irrigation-scheduling was then solved, using the developed crop-water-production function.

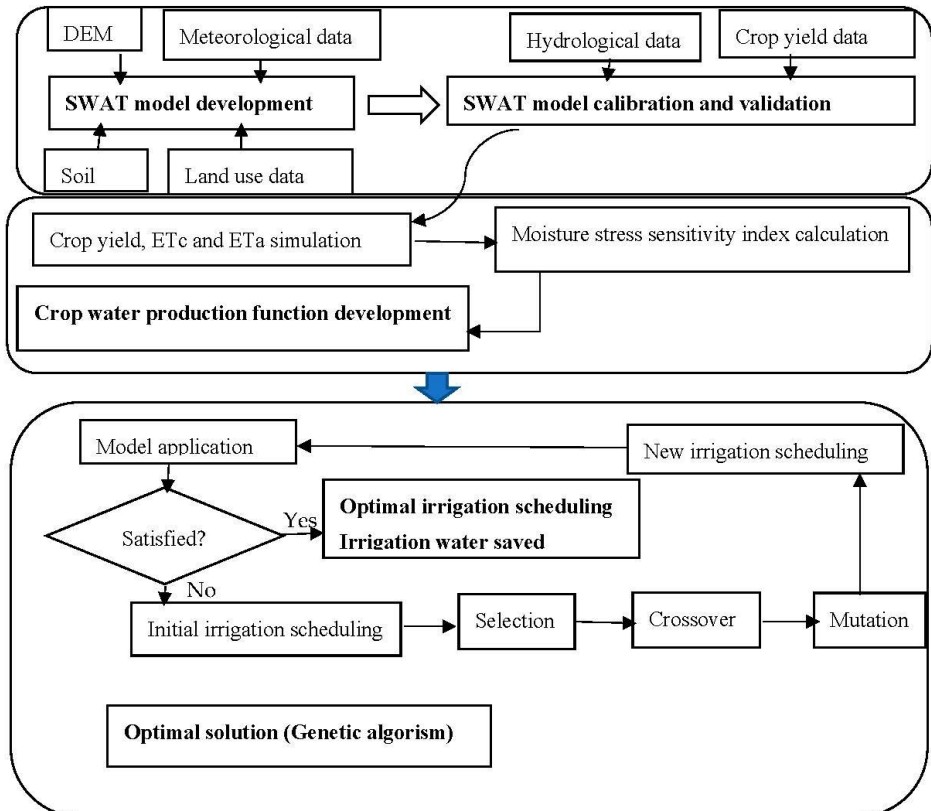

**Figure 3.** Diagrammatic representation of the study.

## 3. Result

### 3.1. SWAT Model Performance

The SWAT model was calibrated and validated using 18 years (1991–2008) of monthly stream-flow data from the Bilate gauging station, based on the data that was available. Of these, two years (1991–1992) of the data were allocated to model warm-up; ten years (1993–2002) of the data were used for calibration; and six years (2003–2008) of the data were adopted for validation. One-at-a-time (OAT) and global sensitivity-analysis methods were applied to identify the most sensitive parameters. The parameters with the smaller *p*-value and the absolute value of the larger t-stat value were nominated for further calibration and validation of the model. The most sensitive parameters were the curve number (CN2), groundwater-recession factor (ALPHA_BF), time taken for water to exit from beneath the root zone (GW_DELAY), threshold depth of water in the shallow aquifer required for return flow to occur (GWQMN), soil-evaporation compensation factor (ESCO), available water capacity of the soil layer (SOIL_AWC), soil moist-bulk density (SOL_BD), Manning's n value for overland flow (OV_N), average slope-length of the watershed (SLSUBBSN), and deep-aquifer percolation fraction (RCHRG_DP). After simulation, the performance of the model was evaluated, using performance indicators. According to the performance indicators, the agreement between measured- and simulated-stream-flow data was good. The timings of flow events (peaks and valleys) were also well estimated (Figure 4). The statistical values indicated that for the calibration period, the values of $R^2$ and $E_{NS}$ were 0.72, and the value of RSR was 0.53, while in the validation period, the values of $R^2$, $E_{NS}$, and RSR were 0.72, 0.65, and 0.59, respectively. Based on Moriasi et al. (2015) criteria, the model showed good performance in the study area.

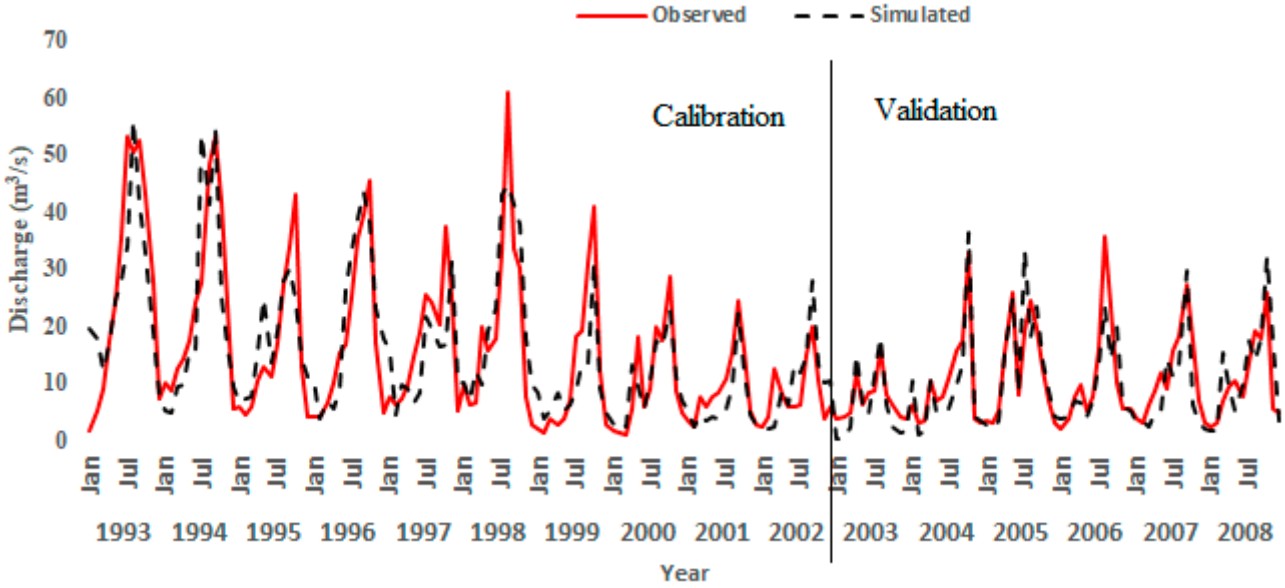

**Figure 4.** Monthly observed- and simulated-stream-flow for calibration and validation period.

Crop parameters were also calibrated manually, using annual average-yield data collected from different districts in the watershed. The fraction of leaf-area index, harvest index, and parameters related to growing season-length had more influence on yield and crop evapotranspiration during simulation. The identified crop parameters and their values before and after calibration are presented in Table 3.

**Table 3.** Adjusted crop-parameters.

| Parameter | Parameter Description | Potato | | Wheat | |
|---|---|---|---|---|---|
| | | Before | After | Before | After |
| BLAI | Maximum leaf-area index | 4 | 4.5 | 4.0 | 4.0 |
| DLAI | Fraction of growing season when leaf area starts declining | 0.6 | 0.6 | 0.8 | 0.8 |
| LAIMX1 | Fraction of BLAI at point 1 | 0.01 | 0.05 | 0.01 | 0.04 |
| LAIMX2 | Fraction of BLAI at point 2 | 0.95 | 0.90 | 0.95 | 0.84 |
| FRGRW1 | Fraction of the plant-growing season at point 1 | 0.10 | 0.15 | 0.15 | 0.10 |
| FRGRW2 | Fraction of the plant-growing season at point 2 | 0.5 | 0.45 | 0.5 | 0.45 |
| HVSTI | Harvest index | 0.95 | 0.90 | 0.4 | 0.35 |

*3.2. The Relationship between $P_e$ and $ET_c$ in the Target Season*

The coupling degree indicates how much the effective precipitation meets the crop water-demand in the growth stages. In this study area, the annual rainfall pattern has bimodal characteristics, with a short rainy season (March–May) and the main rainy season (June–September). Usually, irrigation agriculture is practiced in the area from November to the start of the main rainy season. $ET_o$ in this period is much greater than the rainfall (Figure 5). The coupling degree among $P_e$ and $ET_c$ indicated that $P_e$ in this period could not fulfill the required amount of $ET_c$ for potato and wheat throughout the growing season (Figure 6).

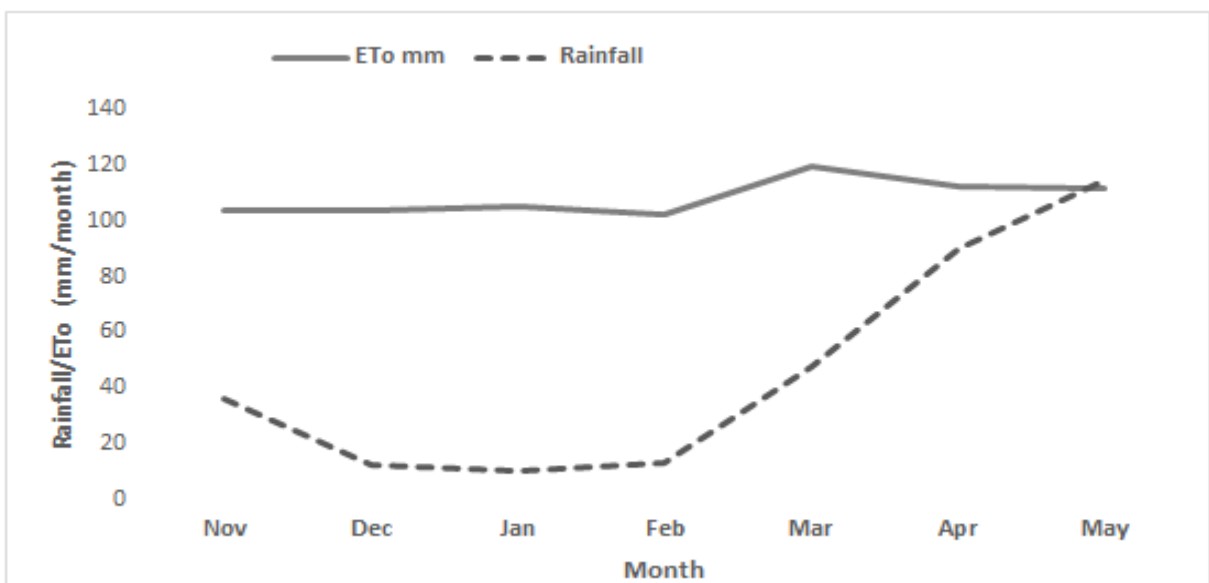

**Figure 5.** Mean monthly $ET_o$ and rainfall in the irrigation season.

*3.3. Statistical Analysis of the Simulated Yield*

The statistical analysis of the SWAT simulated yield indicated that the yield of all deficit-irrigation treatments showed a significant difference from the full irrigation in both crops. However, the significance level varies with the stage of growth at which the deficit was scheduled and the amount of the deficit. Water stress at seedling and maturity stages has less effect on yield than at vegetative and starch-accumulation/grain-filling stages. With the least significant difference (LSD) level of 288.3 and 165.7 for potato and wheat, respectively, the yield difference between the full-irrigation treatment and all deficit-irrigation treatments is presented in Table 4.

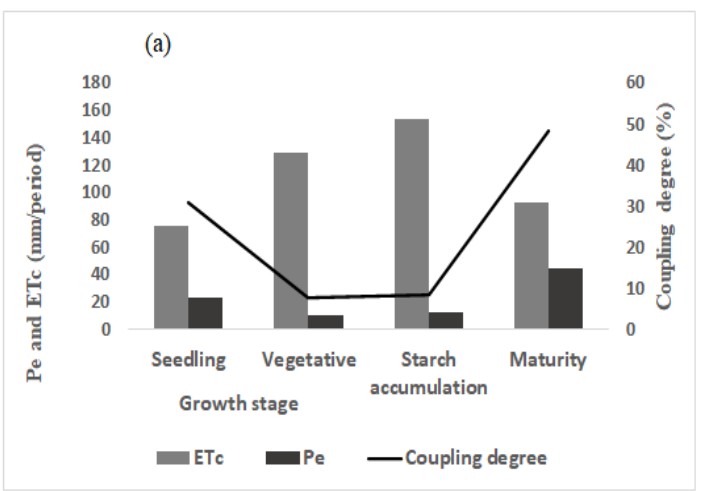 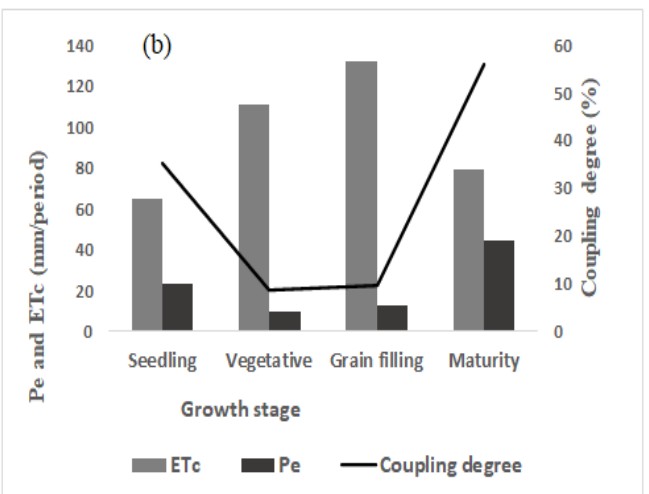

**Figure 6.** Coupling degree between $P_e$ and $ET_c$ in the irrigation season: (**a**) potato, (**b**) wheat.

**Table 4.** Statistical analysis of SWAT simulated yield.

| | Potato | | | Wheat | |
|---|---|---|---|---|---|
| **TRT Rank** | **Yield (kg/ha)** | **% of Yield Reduced** | **TRT Rank** | **Yield (kg/ha)** | **% of Yield Reduced** |
| CK | 8056.609 [a] | - | CK | 4501.07 [a] | - |
| T1 | 7308.935 [b] | 9 | T1 | 4060.017 [b] | 10 |
| T4 | 7252.461 [b] | 10 | T4 | 3986.577 [b] | 11 |
| T7 | 6742.744 [c] | 16 | T7 | 3678.789 [c] | 18 |
| T2 | 5825.637 [d] | 28 | T2 | 3056.719 [d] | 32 |
| T6 | 5576.38 [d,e] | 31 | T3 | 2829.889 [e] | 37 |
| T3 | 5414.225 [e] | 33 | T6 | 2804.272 [e] | 38 |
| T5 | 4193.523 [f] | 48 | T5 | 2086.153 [f] | 54 |
| LSD | 288.3 | | LSD | 165.7 | |

Note(s): There are no significant yield differences between treatments in a column with the same letter at $p < 0.05$.

The Jensen moisture stress-sensitivity index was calculated using the simulated maximum and actual yield and evapotranspiration. The simulated yield and evapotranspiration data were used from HRUs in five representative subbasins based on the agro-ecology of the watershed. The moisture stress-sensitivity index varies across subbasins, particularly at vegetative and starch-accumulation/grain-filling stages (Table 5). In both crops, the moisture stress-sensitivity index at vegetative and starch-accumulation/grain-filling stages is greater than at seedling and maturity stages. The average regional moisture-sensitivity-index for potato is 0.05, 0.28, 0.32, and 0.06 at seedling-, vegetative-, starch-accumulation-, and maturity-stages, respectively, and the regional average moisture-sensitivity-index for wheat is 0.06, 0.36, 0.40, and 0.07 at seedling-, vegetative-, grain-filling-, and maturity-stages, respectively. For both crops, the Jensen crop-water-production function was established, using the calculated moisture stress-sensitivity index.

The Jensen crop-water-production function for potato in the study area:

$$\frac{Y_a}{Y_m} = \left(\frac{ET_{a1}}{ET_{m1}}\right)^{0.05} * \left(\frac{ET_{a2}}{ET_{m2}}\right)^{0.28} * \left(\frac{ET_{a3}}{ET_{m3}}\right)^{0.32} * \left(\frac{ET_{a4}}{ET_{m4}}\right)^{0.06} \tag{12}$$

The Jensen crop-water production function for wheat in the study area:

$$\frac{Y_a}{Y_m} = \left(\frac{ET_{a1}}{ET_{m1}}\right)^{0.06} * \left(\frac{ET_{a2}}{ET_{m2}}\right)^{0.36} * \left(\frac{ET_{a3}}{ET_{m3}}\right)^{0.40} * \left(\frac{ET_{a4}}{ET_{m4}}\right)^{0.07} \tag{13}$$

The developed Jensen crop-water-production function model estimated the relative yield of treatments with an $R^2$ of 0.99 for both crops and root mean square errors (RMSE) of 0.068 and 0.08 for potato and wheat, respectively (Table 6). The Jensen model predicts the relative yield accurately for less-deficit treatments (T1 and T4). For high irrigation-deficit treatments (T5), prediction accuracy is reduced. The average prediction performance of the model is good.

**Table 5.** Moisture stress-sensitivity index in selected subbasins.

| Sub-Basins | Growth Stages of Potato | | | | Growth Stages of Wheat | | | |
|---|---|---|---|---|---|---|---|---|
| | Seedling | Vege.t | Starch ac. | Maturity | Seedling | Vege.t | Grain fill. | Maturity |
| 1 | 0.03 | 0.12 | 0.17 | 0.04 | 0.03 | 0.14 | 0.24 | 0.04 |
| 5 | 0.05 | 0.11 | 0.17 | 0.02 | 0.02 | 0.11 | 0.19 | 0.04 |
| 8 | 0.04 | 0.25 | 0.31 | 0.05 | 0.04 | 0.38 | 0.38 | 0.06 |
| 12 | 0.06 | 0.26 | 0.30 | 0.04 | 0.07 | 0.38 | 0.38 | 0.06 |
| 26 | 0.07 | 0.46 | 0.47 | 0.10 | 0.09 | 0.49 | 0.55 | 0.09 |
| Basin level | 0.05 | 0.28 | 0.32 | 0.06 | 0.06 | 0.36 | 0.40 | 0.07 |

**Table 6.** SWAT-simulated and Jensen-model-predicted relative yield.

| TRT | Potato | | Wheat | |
|---|---|---|---|---|
| | SWAT Simulated Relative Yield | Jensen Predicted Relative Yield | SWAT Simulated Relative Yield | Jensen Predicted Relative Yield |
| 1 | 0.91 | 0.93 | 0.90 | 0.91 |
| 2 | 0.72 | 0.68 | 0.68 | 0.60 |
| 3 | 0.67 | 0.64 | 0.63 | 0.57 |
| 4 | 0.90 | 0.91 | 0.89 | 0.90 |
| 5 | 0.52 | 0.37 | 0.46 | 0.29 |
| 6 | 0.69 | 0.61 | 0.62 | 0.54 |
| 7 | 0.84 | 0.82 | 0.82 | 0.77 |
| $R^2$ | 0.99 | | 0.99 | |
| RMSE | 0.068 | | 0.08 | |

### 3.4. Irrigation-Scheduling Optimization

The seasonal irrigation cycles and developed crop-water-production function were used to optimize irrigation-scheduling in the optimization model. The calculated moisture stress-sensitivity indices were transformed into corresponding irrigation cycles, using the cumulative-sensitivity-index curve (Figure 7) and number of days in each growth stage. In the study area, the average number of days for a full growing season of potato and wheat is 115 and 90, respectively. For potato, 10, 15, 40, 35, and 15 intra-seasonal growth-stage days for establishment, seedling, vegetative, starch accumulation, and maturity, respectively, were considered. Similarly, for wheat, 7, 8, 35, 25, and 15 intra-seasonal growth-stage days for establishment, seedling, vegetative, grain filling, and maturity, respectively, were considered. Taking into account the availability of irrigation water in the study area, a fifteen-day irrigation interval was assumed (Table 7).

The seasonal relative-evapotranspiration of the deficit-irrigation treatments was optimized in between irrigation-cycles, based on the transformed moisture stress-sensitivity index. Considering the length of growing seasons, seven and six seasonal irrigation-cycles were adopted for potato and wheat, respectively. The genetic algorithm toolbox on the MATLAB 2020a platform was used to solve the optimal value of relative evapotranspiration for maximizing relative yield, and results are presented in Figures 8 and 9. The percentage of maximized yield was determined, relative to the yield from prior optimization. The optimization result indicated that the seasonal evapotranspiration of all treatments showed some level of yield maximization, except at T5 and T6. In both crops, the highest yield maximization was attained at T3 and T2 (Figure 10). In the case of potato, the highest yield

maximization was obtained at T3 (25%), followed by T2 (21%). Similarly, for wheat, the highest yield maximization was achieved at T3 (34%), followed by T3 (29%).

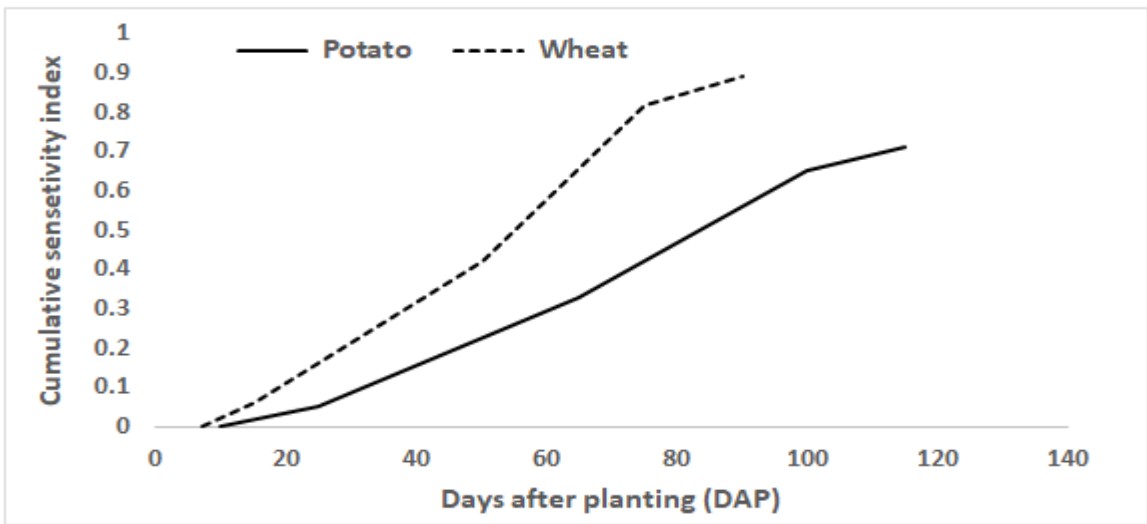

**Figure 7.** Cumulative-sensitivity-index curve.

**Table 7.** Transformed moisture stress-sensitivity index in fifteen-days interval.

| | Potato | | Wheat |
| DAP | Transformed Sensitivity Index | DAP | Transformed Sensitivity Index |
| --- | --- | --- | --- |
| 10 | 0 | 7 | 0 |
| 25 | 0.05 | 15 | 0.06 |
| 40 | 0.105 | 30 | 0.154 |
| 55 | 0.105 | 45 | 0.154 |
| 70 | 0.1157 | 60 | 0.211 |
| 85 | 0.1371 | 75 | 0.24 |
| 100 | 0.1371 | 90 | 0.07 |
| 115 | 0.06 | | |

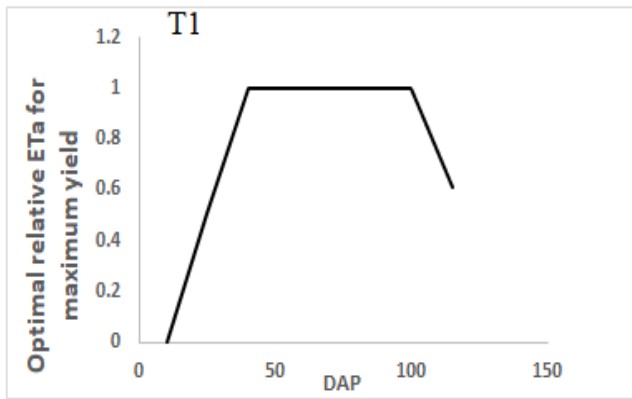

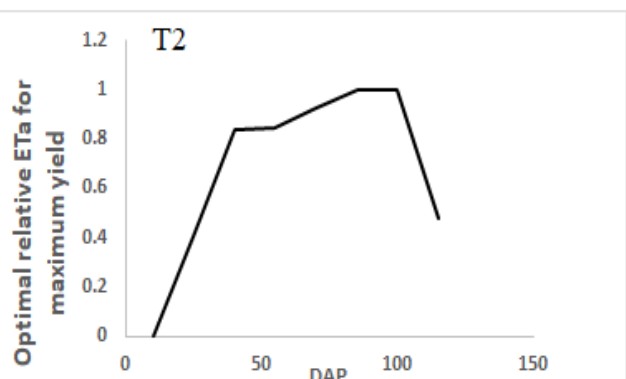

**Figure 8.** *Cont.*

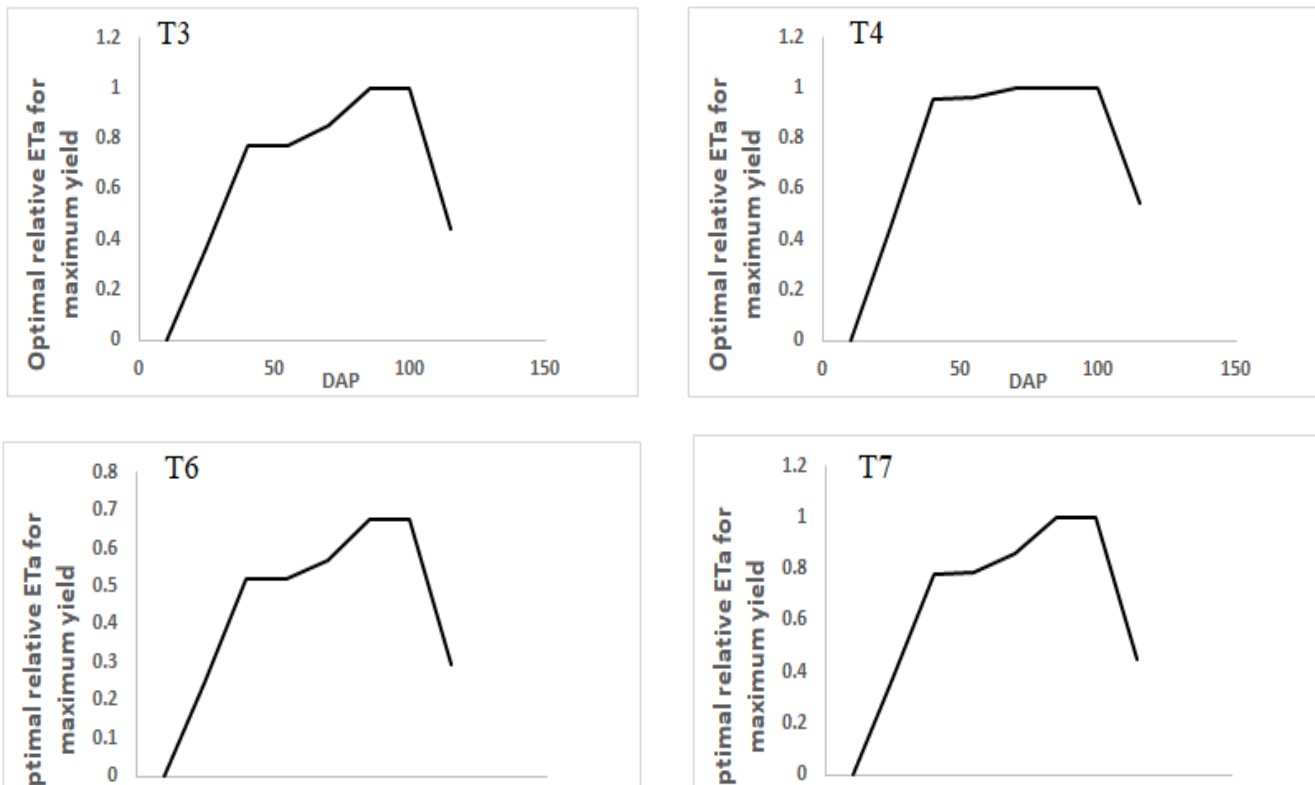

**Figure 8.** Optimal relative ETa for potato under different levels of seasonal-irrigation water.

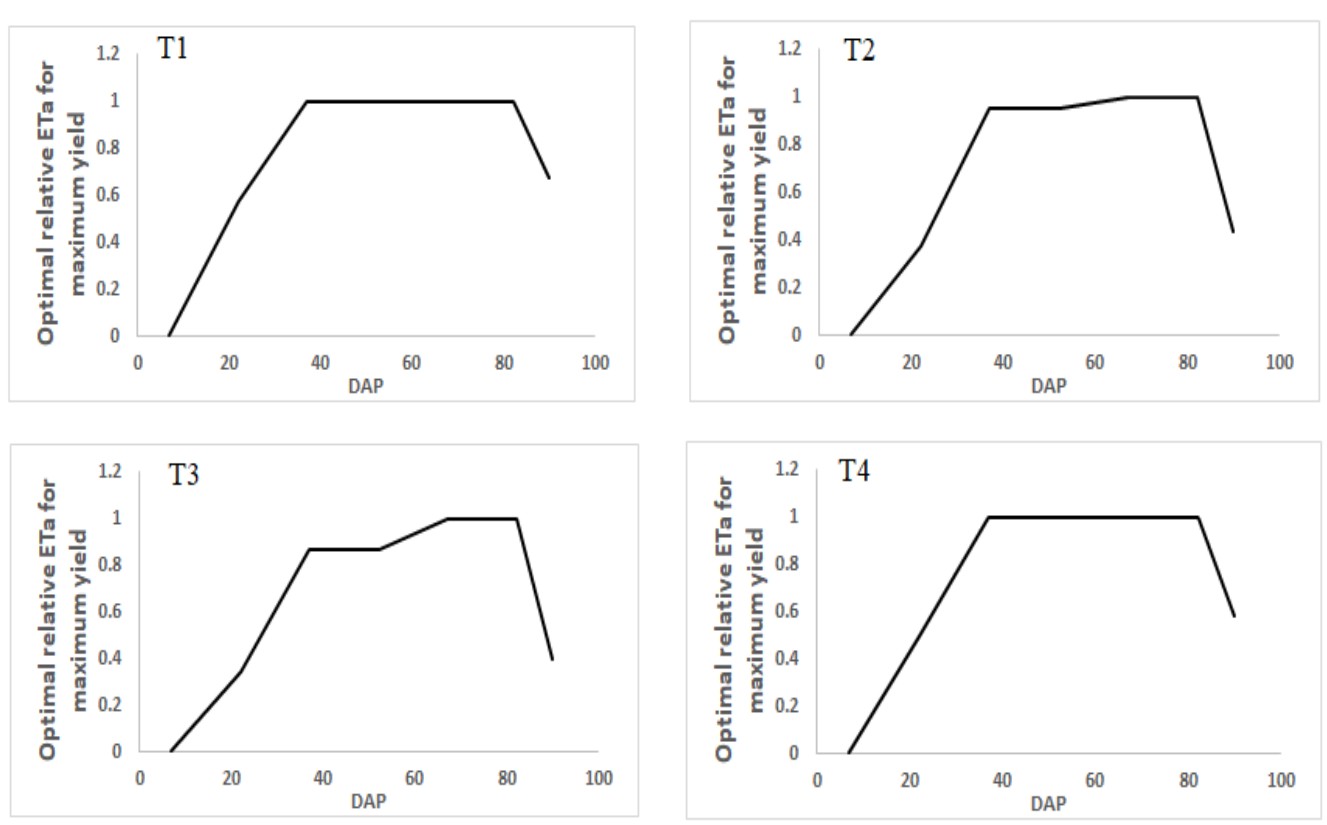

**Figure 9.** *Cont.*

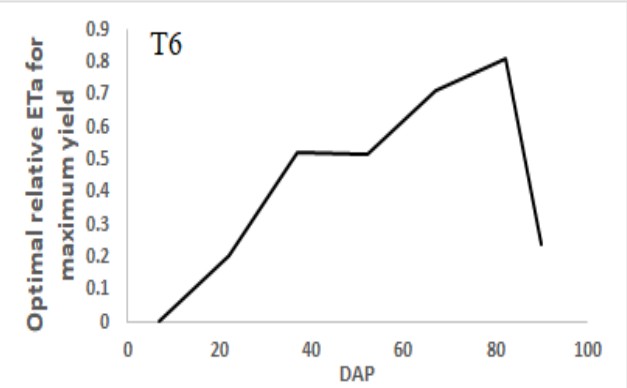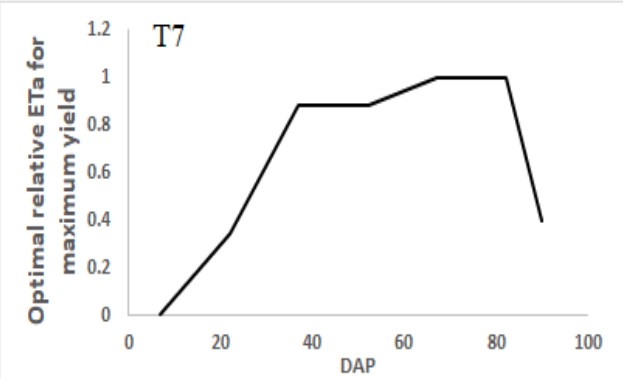

**Figure 9.** Optimal relative ETa for wheat under different levels of seasonal-irrigation water.

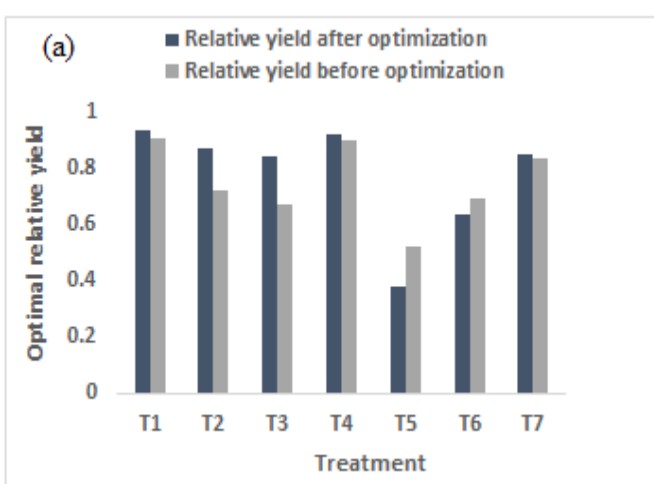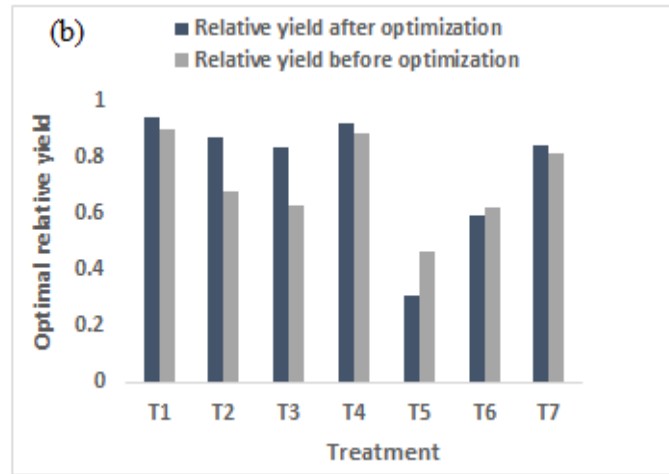

**Figure 10.** Optimal relative yield before and after optimization: (**a**) potato and (**b**) wheat.

In addition, yield maximization was also achieved at T1, T4, and T7 in both crops, but the amount was smaller, compared with T2 and T3. Prior to optimization, the yields of potato and wheat were reduced by 33% and 37%, respectively, at T3 (Table 4), while after optimization, the yields increased by 25% and 34%, respectively. At T2, the yields of potato and wheat were reduced by 28% and 32%, respectively, before optimization; however, the yields increased by 21% and 29%, respectively, after optimization. On the other hand, irrigation-scheduling optimizations at T5 and T6 were unable to maximize yield in both crops. In these treatments, the simulated yield was higher than the yield after optimizing the irrigation schedule.

## 4. Discussion

Irrigation-scheduling optimization is an important strategy to cope with climate change impacts and the shortage of agricultural water-resources [12,52]. In this study, non-deficit- and deficit-irrigation treatments were scheduled in the SWAT model, and yield and evapotranspiration of potato and wheat crops were simulated. The simulated yield and evapotranspiration from each HRU in the selected subbasins were used to compute the Jensen moisture stress-sensitivity index for the two crops. Two groups of deficit-irrigation treatments were used. In the first group, water deficits were scheduled only at a single growth-stage, which allows for distinguishing the most moisture-sensitive growth stages. In the second group, water deficits were triggered at all growth stages, based on the ETc of the specific growth-stages, which is also important for examining the water-stress level and its impact on yield. The computed moisture stress-sensitivity indexes were used to establish the Jensen crop-water-production function model. Irrigation-scheduling

was then optimized, using the developed crop-water-production function for seasonal irrigation-intervals.

The findings indicated that water stress at vegetative- and starch-accumulation/grain-filling stages, lowers production more significantly. These stages of the crop growth-cycle are dominated by tillering and reproduction, and this is when the crop photosynthetic-activity peaks. At this point, water stress will have a more negative impact on vegetative growth and production. Comparable findings from field experiments on wheat by [53] and potato by [51,54] have been reported. The moisture stress-sensitivity index of the two crops was high at the vegetative and starch-accumulation/grain-filling stages and low at the seedling and maturity stages. In fact, at the later stages, the leaf area and canopy size of the crops are relatively small. Therefore, crop water-use during these stages is low, and yield reduction due to moisture stress is less significant. As a result, the majority of the water applied at these stages evaporates from the soil. Similar results have been reported in different parts of the world, such as by [51] for potato and [55] for wheat. However, the magnitude of the moisture stress-sensitivity indexes differs. This variation might be due to differences in local climate conditions and moisture-stress levels. The relative crop yield was estimated using a developed water-production function, which is associated with the moisture-stress-sensitivity index parameter. Compared to the simulated relative yield, the Jensen model accurately predicted the relative yield, with negligible errors. This conclusion is supported by the findings of a maize field-experiment conducted by [44].

The optimization of irrigation-scheduling for crops using moisture stress-sensitivity levels is a practical method for saving irrigation water and reducing associated production costs. In this study, the seasonal relative evapotranspiration of different deficit levels of irrigation water was optimized, to evaluate yield maximization. Since our goal was to maximize yield with deficit irrigation, all maximized yields following optimization were compared to the yield under full irrigation. The optimization result indicated that in both crops, yield maximization was achieved at T3, T2, T1, T4, and T7 (Figure 10). At T3 and T2, the yield was maximized to a greater extent than with other treatments. This was due to the fact that, first, T1 and T4 had lower seasonal deficit-levels than other treatments; second, the deficit at T1 and T4 was scheduled at seedling and maturity growth-stages, respectively. For these reasons, the yield reduction brought on by water stress at T1 and T4 was less significant, in comparison with other deficit treatments. Since T1 and T4 were initially close to optimal (the yield of full irrigation), the amount of maximized relative yield after optimization was less than T3 and T2. Increasing irrigation water-consumption gradually boosts yield, until it reaches the optimum level, after which additional increases in irrigation water would not increase yield but might even slightly reduce it [12]. On the other hand, the seasonal amount of irrigation water-level at T3 and T7 was almost equal in both crops. The simulated yield of T3 was far less than T7. However, after irrigation-scheduling optimization, the relative yield of the two treatments came to be approximately equal. As discussed above, a high-yield reduction was observed when moisture stress was scheduled at the vegetative and starch-accumulation/grain-filling stages (T3 and T2). After optimizing the relative evapotranspiration between irrigation cycles, significant yield maximization was achieved at T3, with the same amount of seasonal-irrigation water. Generally, the results indicated that scheduling the irrigation water for growing seasons based on the moisture stress-sensitivity level of the crops is valuable for saving irrigation water and maximizing the yield of deficit irrigation. In times of water scarcity, it also enables irrigators to determine how much water they need to maintain for the optimal yield. In addition, such kinds of irrigation-scheduling optimization allow a substantial degree of flexibility in planning the irrigation interval, to consider different soil and climatic conditions [46].

This study also revealed that optimizing irrigation-scheduling does not always reflect optimistic results. Optimizing irrigation-scheduling in the case of a high irrigation-water-deficit level may not maximize yield. As it is shown in Figure 10, irrigation-scheduling optimization at T5 (75% deficit throughout the growing season) and T6 (50% deficit through-

out the growing season) was not successful in either crop. This suggests that, for better outcomes, the crop water-requirement level should be considered when optimizing irrigation-scheduling.

## 5. Conclusions

To conserve irrigation water, irrigation-scheduling optimization was developed, using a simulation-optimization model. Following calibration of the sensitive parameters, deficit and non-deficit irrigation treatments were scheduled in the SWAT model, to identify the moisture-stress-sensitive growth-stages of potato and wheat. The crop-water production function of potatoes and wheat in the study area was calculated using the Jensen moisture stress-sensitivity index. Different seasonal deficit-irrigation levels were optimized between seasonal irrigation-cycles for yield maximization. The general conclusions are:

(1) The model can be applied to manage the complicated simulation-optimization irrigation-scheduling problems for wheat and potato, in the study area.

(2) The Jensen moisture stress-sensitivity index indicated that the vegetative and starch-accumulation/grain-filling growth-stages of potato and wheat crops are the most moisture-stress-sensitive stages. Moisture stress at these stages would significantly lower the crop yield.

(3) Optimizing irrigation-scheduling based on growth-stage moisture-stress-sensitivity levels can save up to 25.6% of irrigation water in the study area, with insignificant yield-reduction. Furthermore, optimizing deficit irrigation-scheduling based on moisture stress-sensitivity levels can maximize the yield of potato and wheat by up to 25% and 34%, respectively.

(4) Planning to save irrigation water should be based on the ETc of the crops. That means irrigation-scheduling optimization may not be effective if the seasonal-irrigation water is too low, compared with ETc.

Furthermore, additional water-stress-based optimization experiments are recommended, to expand on the current findings in the study area.

**Author Contributions:** K.W. was responsible for planning the research, data collection and analysis, and writing the first draft. A.H., T.A., S.T. and M.E.-A. supervised the analyses of the results and added their valuable comments to modify the manuscript. All authors have read and agreed to the published version of the manuscript.

**Funding:** This study was funded by Network of Excellence on Land Governance in Africa (NELGA)/ German Academic Exchange Service (DAAD).

**Data Availability Statement:** Data will be available upon request to the corresponding author.

**Acknowledgments:** We wish to thank the Network of Excellence on Land Governance in Africa (NELGA)/German Academic Exchange Service (DAAD). We would like to thank the Ethiopian Ministry of Water and Energy for providing hydrological data of the watershed. We wish to thank the National Meteorological Agency of Ethiopia (NMA) for providing the climatic data of the study area. We are also grateful to the Worabe Agricultural Research Center for providing a vehicle to collect field data. Last, but not least, we would like to thank the Siltie Zone Agricultural Department for providing crop-yield data.

**Conflicts of Interest:** The authors declare no conflict of interest.

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
