# Peer review of "Optimization of Irrigation Scheduling for Improved Irrigation Water Management in Bilate Watershed, Rift Valley, Ethiopia"

_water, doi:10.3390/w14233960_

Round 1
Reviewer 1 Report
The manuscript entitled “Optimization of Irrigation Scheduling for Improved Irrigation Water Management in Bilate Watershed, Rift Valley, Ethiopia” is written and explained appropriately.
General comments:
1. The abstract can be rephrased, exclude out irrelevant content.
2. Kindly revise the spacing amid sentences throughout the manuscript.
3. Mention some latest references in the introduction section as per the requirement.
4. Statistical evaluation of data missing. Can execute the significant difference and incorporate the same in tables and figures.
4. There are also too many long sentences that must be taken care of.
5. Strict guidelines of the journal should be followed precisely in the reference style.
6. There are many typographical and grammatical mistakes in the manuscript that need to be taken care of.
Reviewer 2 Report
this paper showed a combination of models to optimize the irrigation schedule, usually the Jenson were regressed by field data instead of the output of a calibrated and validated model, however eventhough the SWAT was calibrated, only stream flow was calibarate, it has almost nothing to do with the Jenson model since Jenson model needs ETc and yield.
Also, some revisions are needed:
line 10-17 you put too much methods in the abstract part.
line 56-63, this paragraph introduced 2 options of SWAT, however it is not a lectural review, this should belongs to the method part.
water is an international journal , the author introduced the Rift Valley Lakes Basin in the first paragraph of introduction which will narrow the background and reduced the reader`s interests.
ALL the figures and tables were poor in the manuscript which should be reivsed
figure2 what do you mean target season, how long term data?
Line 201 this paper aim to optimize the irrigation schedule, in that way the ETc and Pe were so important, so how do you define Pe, how do you calculate it, this should be given, how is ETc calculate is also so important, the corresponding details should be given, also ETc should be calibrated and validated by field data. did you get Kc from T1 treatment?
Table 2, the experiment should be describe more accurately. for example, how can i control the depth for potato for 20% of ETc.
Table 4, the regression performation of Jenson model shoud be also be mentioned here , for example R2, EF, RMSE.
Figure 9, the author has poor knowledge for ET, if the crop are suffering the water stress, then the ET is not ETc anymore. is it called ETa? please refer to FAO56.
Reviewer 3 Report
This is the review comments on Optimization of Irrigation Scheduling for Improved Irrigation Water Management in Bilate Watershed, Rift Valley, Ethiopia by Webela et al. The overall story is interesting and requires good insights. I recommend the paper publishable after minor revisions.
Specifically, the editing needs to be significantly improved---almost all variable names are in wrong formats that are not consistent with those in the equations.
Round 2
Reviewer 2 Report
Thank you for the revision you have made, now the manuscript looks good.
however, the format and figure resolution needs improving.